# CTC: THE COMPOSITE TASK CHALLENGE FOR COOPERATIVE MULTI-AGENT REINFORCEMENT LEARNING

## ABSTRACT

The critical role of division of labor (DOL) in enhancing cooperation is well-recognized in real-world applications. Consequently, many cooperative multi-agent reinforcement learning (MARL) methods have incorporated DOL mechanisms to improve cooperation among agents. However, the lack of benchmark tasks specifically designed to evaluate and promote DOL and cooperation has limited the effective development and deployment of such mechanisms in cooperative MARL. This gap between current cooperative MARL methods and practical applications underscores the need for evaluation tasks that explicitly require DOL and cooperation. To address this gap, we propose the **C**omposite **T**asks **C**hallenge (**CTC**) — a suite of tasks explicitly designed to require both DOL and cooperation for successful task completion. The CTC tasks are constructed based on two core design principles: 1) DOL is a necessary condition for task success; 2) Failure in any atomic subtask results in failure of the overall task. The first principle emphasizes the necessity of DOL, while the second enforces the importance of cooperation, making both components essential for success in CTC tasks. We evaluate nine representative cooperative MARL methods on the proposed CTC tasks. Experimental results show that all methods consistently achieve zero test winning rates across all CTC tasks, highlighting the challenge of CTC tasks and the limitations of current methods. To facilitate future research, we also introduce a guiding solution and achieves non-zero test winning rates on all tasks, thereby demonstrating the solvability of the CTC tasks. However, the performance of this guiding solution remains suboptimal, further underscoring the value of CTC tasks as a challenging and meaningful testbed for advancing cooperative MARL research.

## 1 INTRODUCTION

Division of labor (DOL) is a fundamental organizing principle that enhances efficiency across both the biological world and human society. In the biological world, DOL manifests in various forms, such as the specialized castes in social insects Hölldobler & Wilson (2009), cellular differentiation in multicellular organisms, and the functional specialization observed in colonial marine invertebrates Dunn & Wagner (2006), and even bacteria Crespi (2001). In human society, DOL has been a cornerstone of economic and technological progress since Adam Smith's seminal work "The Wealth of Nations" Smith (2002), and it is still ubiquitous in modern production systems.

Given its pervasive role in fostering cooperation and efficiency, DOL presents a natural paradigm for advancing cooperative policy learning in multi-agent reinforcement learning (MARL). Accordingly, many MARL methods have incorporated DOL into their algorithmic frameworks, typically through one of three main paradigms: policy diversity, agent grouping, and hierarchical MARL. Policy diversity Jiang & Lu (2021); Mahajan et al. (2019); Li et al. (2021) seeks to address the convergence to similar behaviors often observed in parameter-sharing architectures by encouraging agents to adopt distinct behavioral policies. This implicit specialization leads to emergent DOL, where agents develop complementary behaviors tailored to different subtasks. Agent grouping formalizes DOL by partitioning agents into functionally distinct subgroups. Group membership may be determined based on predefined roles Wang et al. (2020a), sub-goals Christianos et al. (2021), task structures Yang et al. (2022), or intrinsic agent capabilities Christianos et al. (2021). Agents within a group typically share a policy, while differentiation between groups enables specialized execution

across subtasks. Hierarchical MARL architectures implement DOL via multi-level task decomposition. The high-level controller manages task decomposition and subtask assignment, while the low-level controllers focus on executing these subtasks. Such architectures align with DOL principles by facilitating systematic subtask specialization, whether through joint action space partitioning Wang et al. (2020b), learned subtask representations Yang et al. (2022), or classification-based task selectors Li et al. (2024). Despite the theoretical integration of DOL in these methods, their practical effectiveness is constrained by a critical shortcoming: the lack of benchmark tasks that explicitly require and reward DOL. More details are available in Sec. A.1.1.

Existing MARL testbeds often only implicitly involve DOL, suffering from two key limitations: 1) DOL is not strictly necessary to complete the task; 2) failure in a single subtask does not directly cause task failure. As a result, agents can succeed without completing all subtasks, undermining the evaluation and optimization of the policies aiming for DOL and cooperation. For example, in the Multi-Agent Particle Environment (MPE) Lowe et al. (2017); Mordatch & Abbeel (2017), while tasks such as **cooperative communication** and **cooperative navigation** involve DOL, the former is limited by predefined roles, and the latter does not penalize partial subtask failure. In Level-Based Foraging (LBF) Christianos et al. (2020); Papoudakis et al. (2021), both DOL and non-DOL policies are viable, rendering DOL optional. Similarly, in multi-robot warehouse environments (RWARE) Papoudakis et al. (2021), DOL is helpful but not critical, as the successful subtasks can compensate for the failed subtasks. Tasks in StarCraft Multi-Agent Challenge (SMAC) Samvelyan et al. (2019b), SMACv2 Ellis et al. (2022), and GRF Kurach et al. (2020) also allow task success through shared policies like focus firing or solo ball control, thus minimizing the need for explicit DOL. Overcooked Carroll et al. (2019) emphasizes sequential decision-making, where DOL is necessary for completing the task. Existing testbeds are insufficient for benchmarking and advancing DOL in cooperative MARL, as they do not explicitly require or evaluate DOL mechanisms. More details are available in Sec. A.1.2.

To advance research of DOL and cooperation in cooperative MARL, we propose the **C**omposite **T**asks **C**hallenge (**CTC**)—a suite of benchmark tasks designed to explicitly require and reward DOL to form successful cooperation. We provide two design principles to ensure that CTC tasks have clear requirements for DOL: 1) DOL is a necessary condition for task success. 2) Failure in any atomic subtask results in failure of the overall task. The first principle emphasizes the necessity of DOL, while the second enforces the importance of cooperation, making both components essential for success in CTC tasks. Considering the diversity and complexity of real-world scenarios, we also incorporate **asymmetry** and **heterogeneity** to increase the realism of CTC tasks. These designs make the CTC tasks more representative of real-world scenarios and highlight the need for sophisticated DOL mechanisms to form successful cooperation. To evaluate existing methods under the CTC tasks, we select 8 representative cooperative MARL methods spanning the three paradigms (policy diversity, agent grouping, and hierarchical MARL) and a classic methods QMIX Rashid et al. (2018); Hu et al. (2021) as baselines. Our experiments reveal that all baselines struggle to succeed on CTC tasks, with all of them achieving zero test winning rates. This highlights a significant gap between the theoretical promise of baselines and their practical execution. Therefore, we propose a guiding solution for the CTC tasks to facilitate further research. This guiding solution includes a rule-based external reward (RER) and an extension of QMIX (e-QMIX). RER enables e-QMIX to achieve non-zero test winning rates across all CTC tasks, demonstrating the solvability of these tasks. Thus, this guiding solution can serve as a valuable guideline for future research. In addition, RER is tailored to CTC tasks implemented in the specific environment and may not generalize to CTC tasks instantiated in other environments, and its effectiveness has not reached an acceptable level. Therefore, further research is still needed. These findings highlight both the inherent challenge of the CTC tasks and their utility in evaluating the capacity of MARL methods to learn reasonable and effective DOL policies to form successful cooperation. This makes CTC tasks facilitate the development of more sophisticated and practical methods in the field of cooperative MARL. Our contributions can be summarized as follows: 1) Testbed: We introduce the CTC tasks, a novel testbed comprising solvable and practical tasks with explicit requirements for DOL, aimed at bridging the gap between MARL methods and real-world applications. 2) Benchmark: We evaluate 9 representative cooperative MARL methods on the CTC tasks, demonstrating their current limitations in effectively leveraging DOL and cooperation. 3) Guiding Solution: We propose a guiding solution to facilitate progress on CTC tasks, providing evidence that the tasks are both challenging and solvable, thus serving as a valuable guideline for future research.

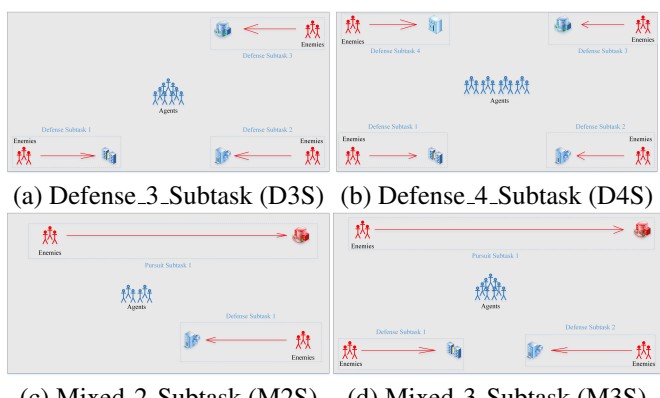

(a) Defense_3_Subtask (D3S)  (b) Defense_4_Subtask (D4S)

(c) Mixed_2_Subtask (M2S)  (d) Mixed_3_Subtask (M3S)

Figure 1: The proposed CTC tasks.

## 2 CTC TASKS

We design the CTC tasks around two core principles: (1) DOL must be a necessary condition for task completion, and (2) the failure of any individual atomic subtask results in overall task failure. The first principle highlights the necessity of DOL, while the second underscores the importance of cooperation, thereby making both components essential for success in CTC tasks. To satisfy **Principle 1**, each CTC task is constructed as a composition of multiple atomic subtasks, where successful completion of all subtasks is required to complete the CTC task. This design also ensures the **decomposability** of CTC tasks, allowing them to be separated into disjoint atomic subtasks. To satisfy **Principle 2**, we define task failure as occurring whenever any atomic subtask is not completed. Furthermore, symmetry between subtasks and agent homogeneity is limited to reflecting simple real-world scenarios, resulting in limited simulation for practical applications. To better capture the diversity and complexity of real-world cooperative tasks, we incorporate both **asymmetry** and **heterogeneity** into the design of tasks. **Asymmetry** is introduced in two ways: by including multiple subtasks of the same type with varying configurations, and by combining subtasks of different types. **Heterogeneity** is introduced through the deployment of agents with diverse types. These design choices not only ensure that CTC tasks more accurately reflect real-world cooperative challenges but also reinforce the necessity of effective DOL and cooperation. **Importantly, all atomic subtasks in a CTC task are initiated simultaneously and separated spatially, preventing any single agent from completing multiple subtasks at the same time. Both agents and enemies are composed of the same types and quantities of units in all CTC tasks. This ensures that performance differences arise from the challenges of the CTC tasks rather than imbalances in agent-enemy capabilities, allowing MARL methods to focus on learning effective DOL policies to form successful cooperation.**

### 2.1 ATOMIC SUBTASKS

Atomic subtasks serve as the foundational components of the CTC benchmark, facilitating the implementation of its two core design principles while also supporting asymmetry and heterogeneity in task structure. To realize these objectives, we introduce two fundamental atomic subtasks: the defense subtask and the pursuit subtask.

In the defense subtask (bottom of Fig. 1(c)), a group of enemy units (red) attempts to occupy a designated base building (blue). The distance between the enemy units' starting point and the base building is set to 7. The subtask is deemed a failure if any enemy unit successfully occupies the base building, while success is achieved if all enemy units are eliminated. In the pursuit subtask (top of Fig. 1(c)), enemy units (red, left) attempt to retreat to their own base building (red, right), located 21 away from their starting position. As in the defense subtask, failure occurs if any enemy unit reaches its base building, while success requires the complete elimination of all enemy units.

Although the pursuit and defense subtasks share structural similarities, they **differ substantially in enemy behavior**. In the pursuit subtask, enemy units do not retaliate when attacked and instead

persistently advance toward their base building. By contrast, in the defense subtask, enemy units engage in combat when attacked. This distinction effectively makes the pursuit subtask **higher priority**: agents must eliminate the retreating enemy units within 21 steps of subtask initiation to avoid failure. In the defense subtask, enemy units occupy the base building only after defeating nearby agents within their attack range. If no agents are within range, they capture the base building 7 steps after the subtask begins. Thus, the pursuit subtask is inherently time-limited, whereas the defense subtask is contingent on combat outcomes.

## 2.2 CTC Implementation

We implement the CTC tasks within SMAC Samvelyan et al. (2019a) for two primary reasons. First, SMAC is the most widely adopted benchmark in cooperative MARL research. By building on SMAC's foundational settings while adhering to the design principles of CTC, we can focus on task construction without being constrained by low-level implementation details. Second, leveraging SMAC ensures that CTC tasks can be seamlessly integrated into most existing MARL methods without requiring modifications to the original source code, thereby substantially reducing the implementation burden for researchers. On this basis, we construct four CTC tasks. Fig. 1 illustrates the spatial configuration of each subtask, and Table 2 details the type of each subtask.

The CTC tasks preserve all of SMAC's original settings, with additional rules introduced solely to enforce the CTC design principles. In all CTC tasks, task success requires the completion of all atomic subtasks, satisfying **Principle 1**, while task failure results from the failure of any atomic subtask, thereby fulfilling **Principle 2**. The two tasks illustrated in Fig. 1(a–b) demonstrate **asymmetry** through the use of atomic subtasks of the same type but with different configurations. Their configurations vary in the number and type of enemy units across atomic subtasks, introducing asymmetry in atomic subtasks within the same task. In addition, the number of atomic subtasks in these two tasks increases progressively (from 3 to 4), thereby gradually increasing the complexity and difficulty of the tasks and establishing a smooth evaluation route. In general, a greater number of atomic subtasks not only elevates task complexity but also more accurately reflects the demands of real-world cooperative scenarios. Methods capable of managing a wider range and larger number of subtasks demonstrate higher applicability and improved scalability, both in terms of task diversity and multi-agent cooperation. The two tasks in Fig. 1(d–e) exemplify **asymmetry** through the composition of different types of atomic subtasks. In practical applications, atomic subtasks are rarely homogeneous, and the degree of heterogeneity directly impacts task complexity and difficulty. Tasks composed of similar subtasks are generally easier than those involving dissimilar ones. Task Mixed_2_Subtask (Fig. 1(c)) includes one defense subtask and one pursuit subtask, representing a balanced composition of dissimilar atomic subtasks. Task Mixed_3_Subtask (Fig. 1(d)) comprises one pursuit and two defense subtasks, introducing an imbalanced proportion of atomic subtask types. These variations allow CTC to more accurately reflect the complexity and irregularity found in real-world cooperative scenarios. To further introduce **heterogeneity**, we utilize the three Terran unit types from StarCraft II—Marine, Marauder, and Medivac—each with distinct capabilities. Marine is light infantry with fast attack speed but relatively low health and damage. Marauder, in contrast, has higher health and armor, and deals more damage but attacks more slowly. Medivac is a non-combat unit with healing capabilities, crucial for sustaining teammates during engagements. Table 1 demonstrates the number and types of agents and enemies of each CTC task. The heterogeneous nature of the agents imposes additional demands on MARL methods, as successful cooperation must consider both task assignment and agent capabilities. This design choice enhances the practical relevance of CTC tasks for evaluating cooperative MARL methods.

## 3 Benchmark

In this section, we present experimental evaluations to assess the performance of cooperative MARL methods on CTC tasks. To evaluate the effectiveness of current state-of-the-art (SOTA) methods on CTC tasks, we select eight representative MARL methods spanning three key categories: policy diversity, agent grouping, and hierarchical MARL. The selected methods are: EOI Jiang & Lu (2021), DCC Li et al. (2024), CDS Li et al. (2021), RODE Wang et al. (2020b), ROMA Wang et al. (2020a), GoMARL Zang et al. (2024), LDSA Yang et al. (2022), and HSD Yang et al. (2019). A more detailed description of baselines is available in Sec. A.1.1. Additionally, we include the classic

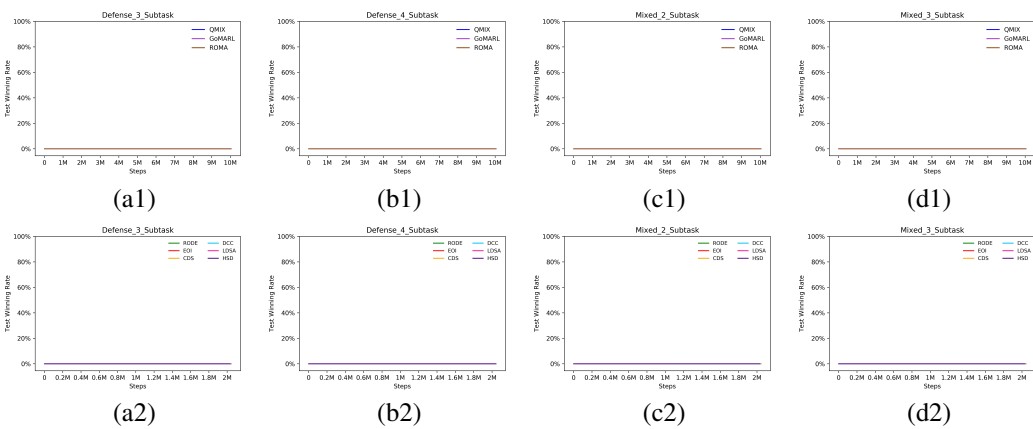

Figure 2: Performance of baselines on CTC tasks.

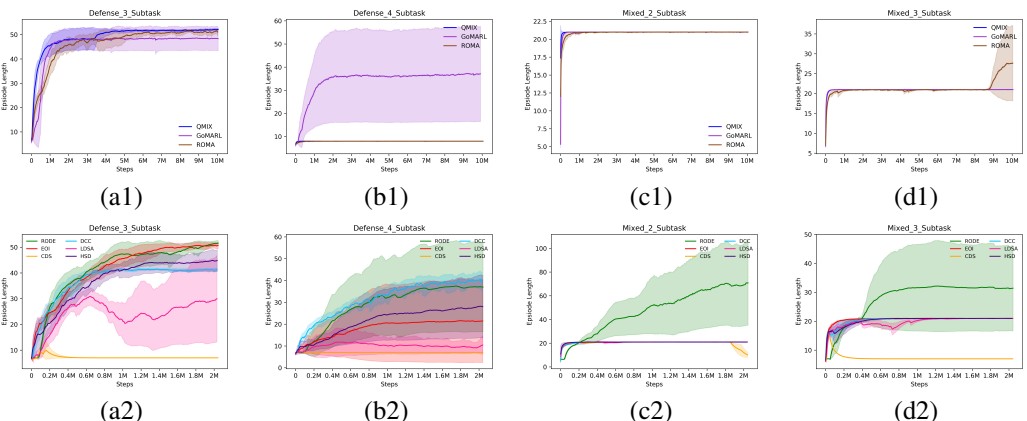

Figure 3: Episode length of baselines on CTC tasks.

cooperative MARL method QMIX Rashid et al. (2018); Hu et al. (2021) as a baseline. We utilize the official implementations of each method and retain their best hyperparameter configurations (Table 5) to ensure fair comparisons. More details about the experimental setting are available in Sec. A.4. Notably, all selected baselines are implemented using the PyMARL framework[1], though they differ in execution strategies. Specifically, QMIX, GoMARL, and ROMA employ parallel runners, executing multiple environments concurrently to collect data. In contrast, the remaining methods utilize episode runners, operating one environment at a time. Due to this divergence, the interpretation of the global step—used to track the number of environment interactions—differs across implementations. To address this inconsistency, we categorize the methods into two groups based on runner type when presenting performance plots. The primary evaluation metric is the test winning rate. We conduct 32 evaluation episodes without exploration and report both the mean and standard deviation of the winning rates across five independent random seeds.

## 3.1 PERFORMANCES

As shown in Fig. 2, **all baselines achieve a test winning rate of 0 on all CTC tasks**. Consequently, direct comparison of test winning rates is meaningless. Instead, we report the mean episode length and its standard deviation for each baseline as a proxy in Fig. 3. Notably, episode length provides insight into how well a method handles specific atomic subtasks: 1) An episode length of less than 7 suggests that at least one defense subtask fails. 2) A length between 7 and 21 indicates that

---

[1]https://github.com/oxwhirl/pymarl

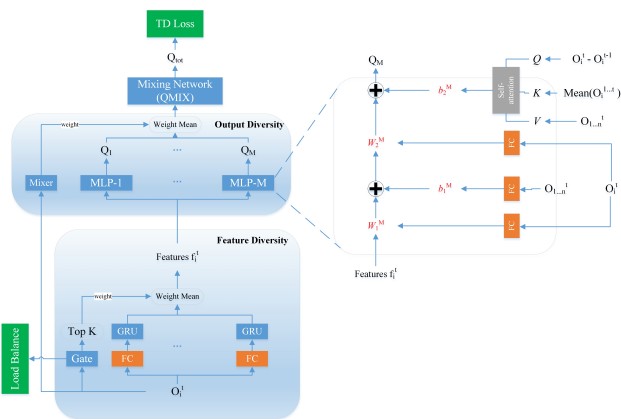

Figure 4: The policy network of extended QMIX.

the pursuit subtask fails. 3) A length greater than 21 implies successful completion of the pursuit subtask, but it does not mean the defense subtask is completed.

In **Defense_3_Subtask**, CDS maintains a constant episode length of 7 (Fig. 3(a2)), indicating that it fails at least one defense subtask and fails to learn a DOL policy. LDSA achieves episode lengths exceeding 7 but shows a downward trend over time, suggesting an awareness of DOL without learning a sufficiently effective DOL policy to ensure task completion. In contrast, other baselines exhibit steadily increasing episode lengths that eventually converge to values significantly higher than 7 (Fig. 3(a1,a2)), indicating stronger DOL learning capacity than CDS and LDSA. A similar trend is observed in **Defense_4_Subtask** (Fig. 3(b1,b2)). However, one notable distinction is that both QMIX and ROMA exhibit stable episode lengths at 7, implying that they fail to learn any effective DOL policy throughout training. Furthermore, the converged episode lengths of most baselines decrease from Defense_3_Subtask to Defense_4_Subtask, aligning with the intentional increase in task complexity by design. In **Mixed_2_Subtask**, all baselines except RODE maintain an episode length of 21 (Fig. 3(c1,c2)), suggesting that they fail to complete the pursuit subtask. RODE shows a consistent increase in episode length, exceeding 60 by the end of training. Nevertheless, its persistent test winning rate of 0 indicates failure in the defense subtask. Upon analysis, this failure stems from the agent assigned to the pursuit subtask remaining idle and wandering after completing its objective—an issue we refer to as the **wandering issue**. We visualize the example of the wandering issue in Fig. 6. In **Mixed_3_Subtask**, the episode length of CDS initially increases to approximately 18 but later declines and stabilizes at 7, indicating instability and ineffectiveness in learning DOL policies. RODE again demonstrates a steadily rising episode length, though the final convergence value is lower than in Mixed_2_Subtask. ROMA exhibits stable episode lengths of 21 during early training but a sharp increase in later stages, suggesting eventual learning of a policy to complete the pursuit subtask. However, its test winning rate remains at 0, indicating that its learned policy is insufficient for ensuring all subtasks are completed. The episode lengths of other baselines remain stable at 21, underscoring their inability to complete the pursuit subtask (Fig. 3(d1,d2)).

In summary, our findings reveal that **all baselines exhibit some degree of DOL capability across all or a subset of the CTC tasks. However, none of the baselines can learn an effective DOL policy to form successful cooperation, as evidenced by their inability to complete the tasks.** These results underscore that CTC tasks pose significant challenges for these cooperative MARL methods. **Therefore, it's essential to develop a guiding solution for the CTC tasks to facilitate further research, which we introduce in the next section.**

## 4 A GUIDING SOLUTION

The guiding solution serves two primary objectives: (1) to demonstrate the solvability of the CTC tasks, and (2) to provide a guide for future research on DOL and cooperation in cooperative MARL. Due to the high complexity of the CTC tasks, we manually design a rule-based external reward (RER) informed by domain knowledge. To better leverage the external reward signal, we further

extend the policy network of QMIX (e-QMIX), enabling improved performance and more effective learning of DOL and cooperation.

## 4.1 RULE-BASED EXTERNAL REWARD

**Damage Reward** Considering the high priority of the pursuit subtask, the designed reward prioritizes the completion of the pursuit subtask. In CTC tasks, the pursuit subtask is executed through a specialized unit, Medivac, which plays a pivotal role in both the pursuit and defense subtasks. This importance arises from two distinctive behavioral features:

1. Feature 1: When no allied units remain, the Medivac continues to move toward its target point, even when under attack.

2. Feature 2: When accompanied by allied units, the Medivac halts and heals them upon being attacked. Once all allies are defeated, it resumes its path toward the target, again ignoring incoming damage.

Feature 1 makes the Medivac a time-sensitive threat in the pursuit subtask. Failure to intercept and eliminate it within a limited time window results in immediate task failure. Feature 2 introduces a prioritization dynamic: targeting the Medivac early disrupts enemy coordination and increases the likelihood of successful subtask completion. Therefore, prioritizing the defeat of the Medivac is a sound and effective policy across both pursuit and defense subtasks. To incentivize this behavior, we design an extrinsic reward based on the reduction in the Medivac's health over time (damage reward), encouraging agents to actively engage with and eliminate these high-priority targets.

$$r_{\text{damage}}^t = \frac{1}{|\mathcal{M}|} \sum_{m \in \mathcal{M}} \frac{\text{HP}_m^{t-1} - \text{HP}_m^t}{\text{HP}_m^{\max}}, \tag{1}$$

where $\text{HP}_m^{\max}$ denotes the maximum health of Medivac unit $m$, while $\text{HP}_m^{t-1}$ and $\text{HP}_m^t$ represent its health at time steps $t-1$ and $t$, respectively. The set $\mathcal{M}$ contains all Medivac units present in the task, and $|\mathcal{M}|$ is the number of units in $\mathcal{M}$.

**Action Reward** A crucial aspect of cooperation in CTC tasks involves assisting teammates in completing their subtasks after one's own has been completed. However, we frequently observe **wandering issue**, which is a key factor limiting the performance of baselines. To address this problem, we incorporate an auxiliary reward that encourages agents to choose actions that lead to positive outcomes, specifically offensive actions that generate environment-level rewards (e.g., attacking actions). This incentive aims to keep agents engaged in meaningful behaviors throughout the episode, thereby promoting more effective cooperation beyond initial task assignments.

$$r_{\text{action}}^t = \frac{1}{n} \sum_{i=1}^n \mathbb{1}(a_i^t \in A^*), \tag{2}$$

where $n$ represents the total number of agents, and $A^*$ denotes the set of attack actions. The indicator function $\mathbb{1}(a_i^t \in A^*)$ equals to 1 if agent $i$ chooses an attack action at time $t$, and 0 otherwise.

Then, **Rule-based Extrinsic Reward (RER)** is formally defined as follows:

$$r_{\text{RER}}^t = \alpha r_{\text{damage}}^t + \beta r_{\text{action}}^t \tag{3}$$

The coefficients $\alpha$ and $\beta$ control the relative importance of each component in RER. The term $r_{\text{action}}^t$ encourages agents to consistently select attack actions, thereby mitigating the **wandering issue**. Since this encouragement should remain effective throughout training, the coefficient $\beta$ is set to a constant value. On the other hand, $r_{\text{damage}}^t$ guides the agents to prioritize targeting the Medivac units. It facilitates learning appropriate grouping policies and temporal prioritization. However, overly strong emphasis on $r_{\text{damage}}^t$ can overshadow the influence of $r_{\text{action}}^t$, especially in later stages of training. To balance this, the coefficient $\alpha$ is designed to decay exponentially over time, allowing the training to initially focus on prioritization while later reinforcing sustained cooperative behavior. This composite reward structure promotes learning effective and cooperative policies by (1) prioritizing the timely elimination of high-impact targets such as Medivacs, and (2) reducing ineffective behaviors such as wandering. Consequently, it fosters MARL methods to learn DOL and cooperation policies, which are essential to solving the CTC tasks.

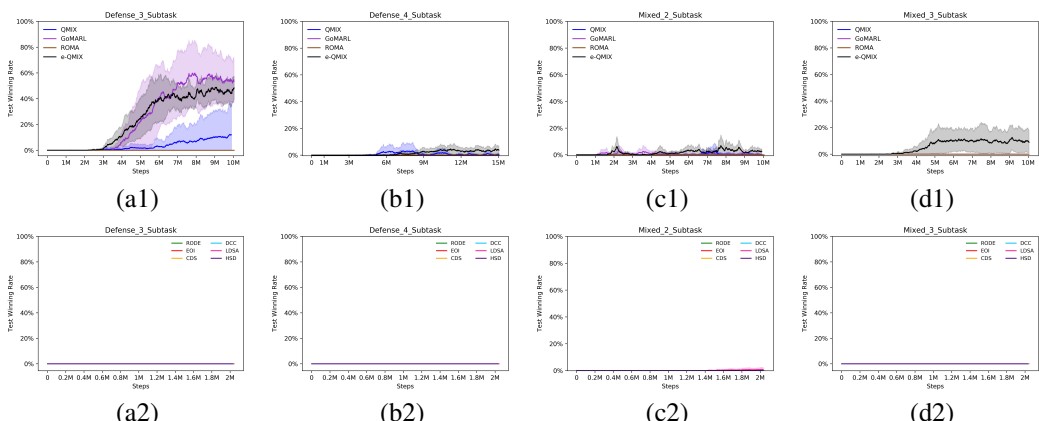

Figure 5: Performance of baselines on CTC tasks with RER.

## 4.2 Extended QMIX

The CTC tasks demand a high degree of policy diversity among agents duo to DOL and cooperation. Theoretically, successful completion of these tasks requires agents to form distinct groups, each assigned to a specific subtask, with the additional expectation that agents provide inter-group support once their own subtasks are completed. **To address this requirement, we extend the policy network of QMIX to enhance its capacity for learning diverse agent behaviors.** We refer to this enhanced version as e-QMIX, and its policy network architecture is illustrated in Fig. 4. The policy network in e-QMIX is composed of two sequential modules: 1) Feature Diversity — designed to enable agents to extract differentiated features even when sharing a policy network. 2) Output Diversity — responsible for producing diverse policies conditioned on those distinct features. Importantly, this extended policy network is still shared across all agents, preserving the parameter-sharing structure of QMIX while enabling richer behavior specialization necessary for solving CTC tasks.

The **feature diversity** is implemented by a sparse mixture of experts (MoE) framework Shazeer et al. (2017). Sparse MoE can promote diversified feature representations which is important for policy diversity. Furthermore, sparse MoE can mitigate the convergence of agent policies that often arises from parameter sharing in MARL Li et al. (2021), which hinders policy diversity. The gating network of sparse MoE produces the score vector, from which the top-$K$ scores are selected to determine the expert weights. This sparse selection ensures that each agent leverages only a subset of available experts, encouraging diverse representations. To prevent load imbalance across experts—which can lead to under-utilization of some experts—we incorporate a balance loss Liu et al. (2024a) to ensure load balance. The loss encourages an even distribution of expert usage across time steps, ensuring that all experts are engaged throughout training.

The **output diversity** component is designed to generate diverse policies for agents, further enhancing behavioral differentiation. To achieve this, we employ a dense MoE architecture Jacobs et al. (1991). The diversity of dense MoE is weaker than that of sparse MoE, but it has the characteristics of smooth output and easy convergence. In particular, each expert consists of a two-layer fully connected (FC) network. The weights of each expert are dynamically generated by separate FC networks conditioned on the agent's local observation. The bias terms are computed as follows: the bias of first layer is generated by a FC that takes as input the concatenated observations of all agents; the bias of second layer is derived using a multi-head attention (MHA) mechanism. The query of MHA is the agent's observation change (i.e., temporal difference in observations). The key is the historical average of the agent's observations. The value is the concatenated observations of all agents. A more detailed description of e-QMIX is provided in Sec. A.2, and an ablation study about e-QMIX is available in Sec. A.7.

## 4.3 PERFORMANCES

The performance of each baseline with RER is presented in Fig. 5. First, e-QMIX consistently achieves non-zero test winning rates across all CTC tasks, thereby demonstrating the solvability of the CTC tasks. Second, RER significantly enhances the performance of both GoMARL and QMIX in certain tasks. For instance, in Defense_3_Subtask (Fig. 5(a1)), RER markedly boosts GoMARL's performance, yielding a maximum test winning rate exceeding 80%. QMIX also benefits from RER, achieving a stable average test winning rate above 10%. Supported by RER, e-QMIX performs comparably to GoMARL in this task. In Defense_4_Subtask (Fig. 5(b1)), the performance gains from RER are less pronounced. Only QMIX shows marginal and unstable improvements, while e-QMIX continues to achieve a stable non-zero average test winning rate. A similar trend is observed in Mixed_2_Subtask (Fig. 5(c1)), where e-QMIX maintains a stable average test winning rate above zero, while QMIX and GoMARL do not exhibit consistent improvements. In Mixed_3_Subtask (Fig. 5(d1)), QMIX and GoMARL maintain average test winning rates near zero. The e-QMIX, however, achieves a modest test winning rate around 10%, albeit with substantial variance across seeds. Interestingly, e-QMIX performs better on Defense_3_Subtask than on Defense_2_Subtask, which may appear counterintuitive given that Defense_3_Subtask is designed to be more difficult. This phenomenon can be attributed to the increased number of agents in Defense_3_Subtask, which offers higher fault tolerance and facilitates more flexible DOL. Nevertheless, RER does not improve ROMA's performance in any of the four tasks. A similar outcome is observed for the baseline using the episode runner (as shown in Fig. 5(a2-d2)), further underscoring the non-universality of RER, as it is not equally effective across all MARL methods. More experimental results are provided in Sec. A.5.

In summary, **with the support of RER, e-QMIX demonstrates the ability to learn a DOL policy and achieve successful cooperation across all CTC tasks**. The strong performance of e-QMIX confirms the solvability of CTC tasks. And we also show the example that e-QMIX solves the wandering issue in Fig. 7. However, RER fails to improve most other baselines, indicating that it is not a universal solution. Moreover, RER is specifically tailored to CTC tasks implemented in SMAC and may not generalize to CTC tasks instantiated in other environments. Therefore, the development of more general and widely applicable solutions for CTC tasks remains an open challenge. Nonetheless, e-QMIX and RER provide valuable insights and lay a foundation for future research on designing effective mechanisms for DOL and cooperation in cooperative MARL.

## 5 CONCLUSION

In this study, we propose the CTC (**C**omposite **T**asks **C**hallenge) tasks, designed to bridge the gap between cooperative MARL methods and practical applications. CTC tasks explicitly require the MARL methods to learn a DOL policy to form successful cooperation, addressing the lack of standardized testbeds that evaluate these core capabilities in existing MARL methods. The CTC tasks are specifically constructed to evaluate and incentivize effective DOL mechanisms, thereby unlocking the full potential of DOL in cooperative MARL. The design of the CTC tasks ensures that DOL are necessary prerequisites for task success, and that the failure of any individual subtask results in the overall task failing. Implemented within the SMAC testbed, the CTC tasks inherit the rationality of SMAC while minimizing the implementation overhead for researchers. We evaluate 9 representative cooperative MARL methods on the CTC tasks. The results reveal that all baselines get test winning rate of 0 on CTC tasks, underscoring the difficulty and discriminative power of the CTC tasks. To further support future research, we introduce a guiding solution including RER and e-QMIX. Experimental results show that the effectiveness of RER is exist but not universal. CTC tasks remain highly challenging for current MARL methods, reinforcing their relevance and value as a testbed for advancing research in cooperative MARL. In summary, the CTC tasks offer a set of solvable yet practical tasks that require DOL and cooperation, providing a valuable platform for pushing the frontiers of cooperative MARL.

## 6 REPRODUCIBILITY STATEMENT

In this section, we outline the measures taken to ensure the reproducibility of our work, which comprises three components: the design and implementation of the CTC tasks, the benchmarking of CTC tasks, and the proposed guiding solution.

For the CTC tasks, the design principles are described in Sec. 2, while implementation details are provided in Sec. 2.2. Additional implementation settings are listed in Sec. A.3. The CTC task files are included in the supplementary materials.

For the benchmark, the experimental setup is presented in Sec. 3, and the configuration of our experimental machine is detailed in Sec. A.4. Links to the source code of each baseline are provided, and the hyperparameters used for the benchmark are summarized in Table 5.

For the guiding solution, the details of RER are discussed in Sec. 4.1, with its implementation available in the e-QMIX source code (parallel_runner.py). The architecture and details of e-QMIX are presented in Sec. A.2, and the corresponding source code is provided in the supplementary materials. The hyperparameters of e-QMIX are listed in Table 4.

## 7 ETHICS STATEMENT

This work focuses on the development and evaluation of cooperative multi-agent reinforcement learning (MARL) algorithms in simulated environments. The proposed CTC benchmark is implemented within the SMAC testbed and involves only synthetic agents and environments. As such, our work does not involve human subjects, personal data, or sensitive demographic attributes, and therefore does not pose risks to individual privacy or safety.

The broader societal impact of this research lies in advancing the understanding of cooperation and division of labor in multi-agent systems, which could contribute to domains such as robotics, logistics, and distributed artificial intelligence. While these applications have the potential to generate positive societal benefits (e.g., improving efficiency in resource allocation and collaboration), we acknowledge that MARL techniques may also be applied in less favorable contexts, such as military or surveillance systems. Our work does not explicitly target such applications, and the benchmark tasks are designed solely for academic research purposes.

We have taken steps to ensure reproducibility by releasing all source code, task definitions, and hyperparameters in the supplementary materials. This transparency supports open research practices and enables the community to critically evaluate and build upon our contributions.

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

# A APPENDIX

## A.1 RELATED WORK

In this section, we first demonstrate that the concept of division of labor (DOL) has been widely integrated into cooperative MARL methods. We then discuss the limitations of the testbeds commonly used in current cooperative MARL research.

### A.1.1 COOPERATIVE MARL METHODS

Specifically, we categorize the cooperative MARL methods that incorporate DOL into three main methods: policy diversity, agent grouping, and hierarchical MARL.

Policy diversity refers to the formation of a cooperative paradigm in which multiple agents, sharing a common goal, adopt diverse policies. By enabling agents to utilize distinct policies, policy diversity implicitly introduces the concept of DOL, where each agent specializes in a specific aspect of the overall task. For instance, EOI Jiang & Lu (2021) suggests that emerging personalities and cooperative behaviors can naturally induce agents to adopt distinct roles and behaviors. In contrast, CDS Li et al. (2021) incorporates diversity into shared policy networks and employs regularization based on information theory to maximize the mutual information between an agent's identity and its trajectory. This method promotes policy diversity while preserving the agents' unique roles. DERE Jiang et al. (2022a) explores the diverse relationships between agents, which can facilitate cooperative policy learning. By introducing prior knowledge to represent these relationships, the method encourages the emergence of specialization, aligning with the concept of multi-agent DOL. SPD Jiang et al. (2022b) proposes an unsupervised MARL method that learns diverse coordination policies for agents without the need for extrinsic rewards. It characterizes agent cooperation using a relational graph, where the varying roles and interactions among agents are reflected, effectively manifesting DOL between agents. In summary, these methods demonstrate how policy diversity can foster specialization and DOL in cooperative MARL, enabling more efficient and coordinated agent behavior.

Grouping agents involves partitioning them into distinct subgroups based on certain similarities, and this structure can be viewed as a form of DOL, where each subgroup specializes in a specific function or task within the broader system. Several cooperative MARL methods adopt this paradigm, employing different concepts to group agents effectively. For example, SEPS Christianos et al. (2021) groups agents based on their abilities and goals. This is achieved by encoding each agent into an embedding space based on their observed trajectories and then applying an unsupervised clustering method to these embeddings. GACG Duan et al. (2024) calculates cooperation needs between agent pairs based on their current observations and captures group dependencies from behavior patterns observed across trajectories. To reinforce group differentiation, it introduces a group distance loss, which increases behavioral differences between groups while minimizing them within groups. QTypeMix Fu et al. (2024a) utilizes prior knowledge of agent types to group agents. Similarly, THGC Jiang et al. (2021) groups agents based on similarities in factors such as location, functionality, and health, while maintaining cognitive consistency within groups through knowledge sharing. VAST Phan et al. (2021) explores value factorization for sub-teams based on the similarity of spatial information or trajectories. ROMA Wang et al. (2020a) implicitly introduces the concept of roles within MARL, facilitating the sharing of learning among agents with similar responsibilities. By ensuring that agents with similar roles share both similar policies and responsibilities, it enables effective coordination. GoMARL Zang et al. (2024) further enhances agent cooperation by empowering subgroups as bridges to model connections between smaller sets of agents, thereby improving overall learning efficiency. It also introduces an automatic grouping mechanism—selecting and removing agents—to dynamically create groups and group action values. In summary, these methods illustrate how grouping agents based on various similarities fosters specialization, enhances coordination, and facilitates the DOL within cooperative MARL systems.

Hierarchical MARL architectures implement DOL via multi-level task decomposition. The high-level controller manages task decomposition and subtask assignment, while the low-level controllers focus on executing these subtasks. Each subtask can be viewed as an agent performing a distinct part of the overall task, thereby embodying the concept of DOL. For example, RODE Wang et al. (2020b) decomposes a multi-agent collaborative task into a set of subtasks, each with a smaller

Table 1: Agents and enemies setting of CTC tasks.

| Tasks | Agents | Subtask 1 | Subtask 2 | Subtask 3 | Subtask 4 |
|---|---|---|---|---|---|
| Defense_3_Subtask | 3 Marine
3 Marauder
3 Medivac | 1 Marauder
1 Medivac | 1 Marine
1 Marauder
1 Medivac | 2 Marine
1 Marauder
1 Medivac | / |
| Defense_4_Subtask | 4 Marine
4 Marauder
4 Medivac | 1 Marine
1 Medivac | 1 Marauder
1 Medivac | 1 Marine
1 Marauder
1 Medivac | 2 Marine
2 Marauder
1 Medivac |
| Mixed_2_Subtask | 2 Marine
2 Marauder
2 Medivac | 1 Medivac | 2 Marine
2 Marauder
1 Medivac | / | / |
| Mixed_3_Subtask | 3 Marine
3 Marauder
3 Medivac | 1 Medivac | 1 Marine
1 Marauder
1 Medivac | 2 Marine
2 Marauder
1 Medivac | / |

Table 2: Subtask type of CTC tasks.

| Tasks | Subtask 1 | Subtask 2 | Subtask 3 | Subtask 4 |
|---|---|---|---|---|
| Defense_3_Subtask | Defense | Defense | Defense | / |
| Defense_4_Subtask | Defense | Defense | Defense | Defense |
| Mixed_2_Subtask | Pursuit | Defense | / | / |
| Mixed_3_Subtask | Pursuit | Defense | Defense | / |

action-observation space. Each subtask is associated with a specific role, and agents within the same role jointly learn a policy to solve the subtask through shared learning. LDSA Yang et al. (2022) learns distinct vectors to represent multiple subtasks and assigns subtask-specific policies to agents based on these vectors, enabling local coordination among agents within each subtask. HSD Yang et al. (2019) focuses on learning distinguishable skills for agents and employs a bi-level policy structure. While it shares a similar objective of fostering local cooperation, it optimizes this objective using a different method. DCC Li et al. (2024) treats subtask decomposition as a fixed number of classification tasks, allowing the direct learning of a subtask selection network to guide agent behavior. In summary, hierarchical MARL methods employ different techniques to decompose complex tasks into subtasks, facilitating specialization and local coordination among agents. This decomposition mirrors the concept of DOL, ensuring that each agent contributes effectively to the completion of the overall task.

Despite the theoretical integration of DOL in these methods, their practical effectiveness is constrained by a critical shortcoming: the lack of benchmark tasks that explicitly require and reward DOL.

### A.1.2 COOPERATIVE MARL TESTBEDS

Multi-Agent Particle Environment (MPE) Lowe et al. (2017); Mordatch & Abbeel (2017) consists of multiple tasks involving the cooperation and competition between agents. All tasks involve particles and landmarks in a continuous two-dimensional environment. Observations consist of high-level feature vectors and agents are receiving dense reward signals. The action space among all tasks and agents is discrete and usually includes five possible actions corresponding to no movement, move right, move left, move up or move down. All experiments in this environment are executed with a maximum episode length of 25, i.e. episodes are terminated after 25 steps and a new episode is started. MPE includes multiple tasks, but only cooperative communication and cooperative navigation involve DOL and cooperation. Cooperative communication consists of two cooperative agents, a speaker and a listener, who are placed in an environment with three landmarks of differing colors. At each episode, the listener must navigate to a landmark of a particular color, and obtains reward based on its distance to the correct landmark. However, while the listener can observe the relative position and color of the landmarks, it does not know which landmark it must navigate to. Conversely, the speaker's observation consists of the correct landmark color, and it can produce a

communication output at each time step which is observed by the listener. Thus, the speaker must learn to output the landmark colour based on the motions of the listener. Obviously, this task requires cooperation between the speaker and the listener, but the roles is determined in advance. From the perspective of MARL methods, this task does not require agents to implement DOL. In cooperative navigation, agents must cooperate through physical actions to reach a set of $L$ landmarks. Agents observe the relative positions of other agents and landmarks, and are collectively rewarded based on the proximity of any agent to each landmark. In other words, the agents have to 'cover' all of the landmarks. Further, the agents occupy significant physical space and are penalized when colliding with each other. The DOL among agents is an effective policy for this task. However, the success or failure of the subtasks after the DOL among agents does not affect the overall success or failure.

The Level-Based Foraging (LBF) environment Christianos et al. (2020); Papoudakis et al. (2021) consists of tasks focusing on the coordination of involved agents. The task for each agent is to navigate the grid-world map and collect items. Each agent and item is assigned a level and items are randomly scattered in the environment. In order to collect an item, agents have to choose a certain action next to the item. However, such collection is only successful if the sum of involved agents' levels is equal or greater than the item level. Agents receive reward equal to the level of the collected item. LBF implicitly includes the concept of DOL among agents, that is, each agent disperses to collect food scattered on the map that is not greater than its own level to improve collection efficiency. However, the DOL is not a necessary condition for completing the task. Even if the DOL among agents is a more efficient policy, the success or failure of the subtasks after DOL does not determine the success or failure of the overall task.

The multi-robot warehouse environment (RWARE) Papoudakis et al. (2021); Christianos et al. (2020) is a set of collaborative, partially observable multi-agent tasks simulating a warehouse operated by robots. Each agent controls a single robot aiming to collect requested shelves. At all times, $N$ shelves are requested and each timestep a request is delivered to the goal location, a new (currently unrequested) shelf is uniformly sampled and added to the list of requests. Agents observe a 3 $\times$ 3 grid including information about potentially close agents, given by their location and rotation, as well as information on surrounding shelves and a list of requests. The action space is discrete and contains of four actions corresponding to turning left or right, moving forward and loading or unloading a shelf. Agents are only rewarded whenever an agent is delivering a requested shelf to a goal position. Therefore, a very specific and long sequence of actions is required to receive any non-zero rewards, making this environment very sparsely rewarded. RWARE implicitly includes the concept of DOL among agents, however, it is not a necessary condition for success. Even if the DOL among agents is a more efficient policy, the success or failure of the subtasks after DOL does not determine the success or failure of the overall task.

StarCraft Multi-Agent Challenge (SMAC) Samvelyan et al. (2019a) is a benchmark suite specifically designed for evaluating Multi-Agent Reinforcement Learning (MARL) methods. Based on the real-time strategy game StarCraft II, SMAC provides a variety of tasks and scenarios in which researchers can test and refine their multi-agent systems. In SMAC tasks, each allied unit is controlled by an RL agent, which can observe the distances, relative positions, unit types, and health of all allied and enemy units within its field of view at each time step. The behavior of enemy units is governed by the built-in rules of the environment. To address certain limitations in SMAC, SMACv2 Ellis et al. (2022) introduces three key modifications: random team compositions, random starting positions, and more realistic field of view and attack range dynamics. These changes encourage agents to focus on understanding their observation space more thoroughly and help prevent the learning of successful open-loop strategies—those that rely solely on the time step for decision-making. Despite these enhancements, SMACv2 and SMAC remain nearly identical in all other aspects. The primary objective for the allied agents in both SMAC and SMACv2 is to eliminate all enemy units within a specified timeframe, with rewards being awarded only when enemy units are eliminated and victory is achieved. Several effective policies have been observed in these environments. For example, a commonly learned policy is to focus fire Li et al. (2023); Liu et al. (2024b; 2023); Mahajan et al. (2019); Yang et al. (2022); Yu et al. (2023); Hu et al. (2023); Shao et al. (2022); Zang et al. (2024) on a single enemy, thereby quickly reducing the number of adversaries and minimizing the damage taken of the agents. Another effective policy involves retreating when an agent's health is low, causing the enemy to switch targets, followed by a counterattack once the agent is no longer a target for any enemy unit. However, neither of these policies involves DOL between agents, as they focus on individual actions rather than collaborative, role-specific tasks.

Google Research Football (GRF) Kurach et al. (2020) is a highly dynamic and complex simulation environment that lacks clearly defined behavior abstractions, making it an ideal testbed for studying multi-agent decision-making and cooperation. The environment adheres to standard football (soccer) rules, including corner kicks, fouls, cards, kick-offs, and offside penalties. Additionally, the physical representation of players is highly realistic, allowing for a diverse range of learning behaviors to be explored, with the option to adjust the difficulty level. In GRF, the model must control a team of agents to compete against an opposing team, with the objective of scoring more goals than the opponent by the end of the match. The environment provides 19 possible actions for the agents, including movement, kicking, and other specialized actions such as dribbling, sliding, and sprinting. GRF also offers several predefined reward signals, such as scoring rewards and a penalty box proximity reward, which encourages attackers to move toward specific locations on the field. In GRF tasks, agents must coordinate their movements, timing, and positioning to organize offensive strategies and seize fleeting opportunities, as rewards are only given for scoring. However, in this environment, all agents are homogeneous and capable of performing all tasks. This means that DOL is not a necessary condition for task completion. For instance, in the academy tasks of GRF, as long as the ball is passed to the agent far away from the defender at the start, the agent dribbling the ball can score alone, and the other agents walking around will not affect the result Fu et al. (2024b); Li et al. (2021); Xu et al. (2023), where DOL is often not a necessary feature for optimal policies.

The Overcooked environment Carroll et al. (2019) is based on the popular video game of the same name Games. The authors implement a simplified version in which the only objects are onions, dishes, and soups. To complete a dish, players must place three onions in a pot, allow them to cook for 20 timesteps, transfer the resulting soup into a dish, and then serve it. Successfully serving a dish yields a shared reward of 20 for all players. The environment supports six possible actions: moving up, down, left, or right; performing no operation (noop); and executing "interact," which performs context-dependent actions based on the tile the agent is facing (e.g., placing an onion on a counter). Each layout contains one or more onion and dish dispensers, which provide an unlimited supply of these items. The majority of layouts are designed to pose either low-level motion coordination challenges or high-level strategy challenges. Accordingly, agents must learn to navigate the map, manipulate and transport objects, and deliver completed dishes to the serving area, all while maintaining awareness of their partner's actions and coordinating effectively. Overcooked thus emphasizes sequential collaboration, requiring multiple agents to work together to complete a shared task. By contrast, our proposed task focuses on the simultaneous execution of multiple subtasks. As a result, methods that perform well in Overcooked may not be well-suited for tasks that require parallel processing and coordination across multiple subtasks.

The environment Overcooked Carroll et al. (2019) is built based on the popular video game Overcooked Games. They implement a simplified version of the environment, in which the only objects are onions, dishes, and soups. Players place 3 onions in a pot, leave them to cook for 20 timesteps, put the resulting soup in a dish, and serve it, giving all players a reward of 20. The six possible actions are: up, down, left, right, noop, and "interact", which does something based on the tile the player is facing, e.g., placing an onion on a counter. Each layout has one or more onion dispensers and dish dispensers, which provide an unlimited supply of onions and dishes, respectively. Most of our layouts were designed to lead to either low-level motion coordination challenges or high-level strategy challenges. Agents should learn how to navigate the map, interact with objects, drop the objects off in the right locations, and finally serve completed dishes to the serving area. All the while, agents should be aware of what their partner is doing and coordinate with them effectively. Overcooked emphasizes sequential decision-making, where DOL is necessary for completing the task. Our proposed task, on the other hand, focuses on the simultaneous processing of multiple subtasks. Methods that perform well on Overcook may not necessarily be suitable for tasks requiring the parallel processing of multiple subtasks.

## A.2 EXTENDED QMIX

The CTC tasks demand a high degree of policy diversity among agents duo to DOL and cooperation. Theoretically, successful completion of these tasks requires agents to form distinct groups, each assigned to specific subtask, with the additional expectation that agents provide inter-group support once their own subtasks are completed. **To address this requirement, we extend the policy network of QMIX to enhance its capacity for learning diverse agent behaviors.** We refer to this

enhanced version as e-QMIX, and its policy network architecture is illustrated in Fig. 4. The policy network in e-QMIX is composed of two sequential modules:

1. Feature Diversity — designed to enable agents to extract differentiated features even when sharing the same network.

2. Output Diversity — responsible for producing diverse action policies conditioned on those distinct features.

Importantly, this extended policy network is still shared across all agents, preserving the parameter-sharing structure of QMIX while enabling richer behavior specialization necessary for solving CTC tasks.

The **feature diversity** is implemented by a sparse mixture of experts (MoE) framework Shazeer et al. (2017). Sparse MoE can promote diversified feature representations which is important for policy diversity. Furthermore, sparse MoE can mitigate the convergence of agent policies that often arises from parameter sharing in multi-agent systems Li et al. (2021), which hinders policy diversity.

$$f_i^t = \sum_{k=1}^{M} g_{i,k}^t E_k(o_i^t), \tag{4}$$

$$g_{i,k}^t = \begin{cases} s_{i,k}^t, & s_{i,k}^t \in \text{TopK}(\{s_{i,j}^t | 1 \leq j \leq M\}), \\ 0, & \text{otherwise}, \end{cases} \tag{5}$$

$$\boldsymbol{s}_i^t = \text{Gate}(o_i^t), \tag{6}$$

Here, $f_i^t$ denotes the feature embedding of agent $i$ at time step $t$, generated as a weighted sum of expert outputs $E_k(o_i^t)$, where $E_k$ is the $k$-th expert network and $o_i^t$ is the agent's local observation. The gating network $\text{Gate}(o_i^t)$ produces the score vector $\boldsymbol{s}_i^t$, from which the top-$K$ scores are selected to determine the expert weights $g_{i,k}^t$. This sparse selection ensures that each agent leverages only a subset of available experts, encouraging diverse representations. To prevent load imbalance across experts—which can lead to under-utilization of some experts—we incorporate a balance loss Liu et al. (2024a):

$$\mathcal{L}_{Bal} = \gamma \sum_{k=1}^{M} c_k P_k, \tag{7}$$

$$c_k = \frac{1}{T} \sum_{t=1}^{T} \infty(s_i^t \in \text{TopK}(\{s_j^t | 1 \leq j \leq M\})), \tag{8}$$

$$P_k = \frac{1}{T} \sum_{t=1}^{T} s_i^t \tag{9}$$

where $\gamma$ is a small weighting hyperparameter, $\infty(\cdot)$ is the indicator function, and $T$ denotes the trajectory length. The loss encourages an even distribution of expert usage across time steps, ensuring that all experts are engaged throughout training.

The **output diversity** component is designed to generate diverse policies for agents, further enhancing behavioral differentiation. To achieve this, we employ a mixture of experts architecture Jacobs et al. (1991), where the final action-value estimate for each agent is computed as:

$$Q_i = \sum_{k=1}^{M} \pi_k E_k'(f_i^t), \tag{10}$$

$$\pi_k = \text{Mixer}(f_i^t), \tag{11}$$

Here, $E_k'(f_i^t)$ denotes the output of the $k$-th expert in the output diversity module, and $\pi_k$ is the corresponding mixing coefficient obtained from a learned mixer network. Each expert $E_k'$ consists of a two-layer fully connected (FC) network. The weights $W_1$ and $W_2$ of the FC layers are dynamically generated by separate FC networks conditioned on the agent's local observation. The bias terms are computed as follows: $b_1$ is generated by a FC that takes as input the concatenated observations of all agents. $b_2$ is derived using a multi-head attention mechanism, where: The query is the agent's

observation change (i.e., temporal difference in observations); The key is the historical average of the agent's observations; The value is the concatenated observations of all agents.

After computing the individual agent Q-values $Q_i$, the subsequent procedure mirrors that of QMIX: each $Q_i$ is passed through a mixing network to form a total Q-value used for computing the TD loss, which then drives backpropagation for policy optimization. This structure promotes both representational and decision-level diversity, essential for effective DOL and cooperation.

## A.3 CTC SETTINGS

SMAC provides a convenient environment for evaluating the effectiveness of MARL algorithms. The simulated StarCraft II environment and carefully designed scenarios require learning rich cooperative behaviours under partial observability, which is a challenging task. The simulated environment also provides additional state information during training, such as information on all the units on the entire map. This is crucial for facilitating algorithms to take full advantage of the centralised training regime and assessing all aspects of MARL methods. SMAC is a qualitatively challenging environment that provides elements of partial observability, challenging dynamics, and high-dimensional observation spaces. By adhering to the design principles of CTC while retaining SMAC's foundational settings, we ensure the implementation rationality of the CTC tasks. Below, we introduce the detailed settings of the CTC tasks. And the type of subtask contained in each CTC task is shown in Table 2.

### A.3.1 SUBTASK CONFIGURATIONS

To better reflect the diversity and complexity of real-world scenarios and increase the challenge of the tasks, we incorporate **asymmetry** and **heterogeneity** into the task and agent designs. We introduce **asymmetry** in two ways: by including multiple atomic subtasks of the same type with varying configurations, and by combining atomic subtasks of different types. **Heterogeneity** is introduced through the deployment of agents with diverse types. The two tasks illustrated in Fig. 1(a–b) demonstrate **asymmetry** through the use of atomic subtasks of the same type but with different configurations. Their configurations vary in the number and type of enemy units across atomic subtasks, introducing asymmetry in atomic subtasks within the same task. The Table 1 lists the agents' and enemies' settings of each CTC task.

### A.3.2 STATE AND OBSERVATIONS

At each timestep, agents receive local observations drawn within their field of view. This encompasses information about the map within a circular area around each unit and with a radius equal to the sight range. The sight range makes the environment partially observable from the standpoint of each agent. Agents can only observe other agents if they are both alive and located within the sight range. Hence, there is no way for agents to distinguish between teammates that are far away from those that are dead.

The feature vector observed by each agent contains the following attributes for both allied and enemy units within the sight range: distance, relative x, relative y, health, shield, and unit_type. Shields serve as an additional source of protection that needs to be removed before any damage can be done to the health of units. All Protos units have shields, which can regenerate if no new damage is dealt. In addition, agents have access to the last actions of allied units that are in the field of view. Lastly, agents can observe the terrain features surrounding them, in particular, the values of eight points at a fixed radius indicating height and walkability.

The global state, which is only available to agents during centralised training, contains information about all units on the map. Specifically, the state vector includes the coordinates of all agents relative to the centre of the map, together with unit features present in the observations. Additionally, the state stores the energy of Medivacs and cooldown of the rest of the allied units, which represents the minimum delay between attacks. Finally, the last actions of all agents are attached to the central state.

All features, both in the state as well as in the observations of individual agents, are normalised by their maximum values. The sight range is set to nine for all agents.

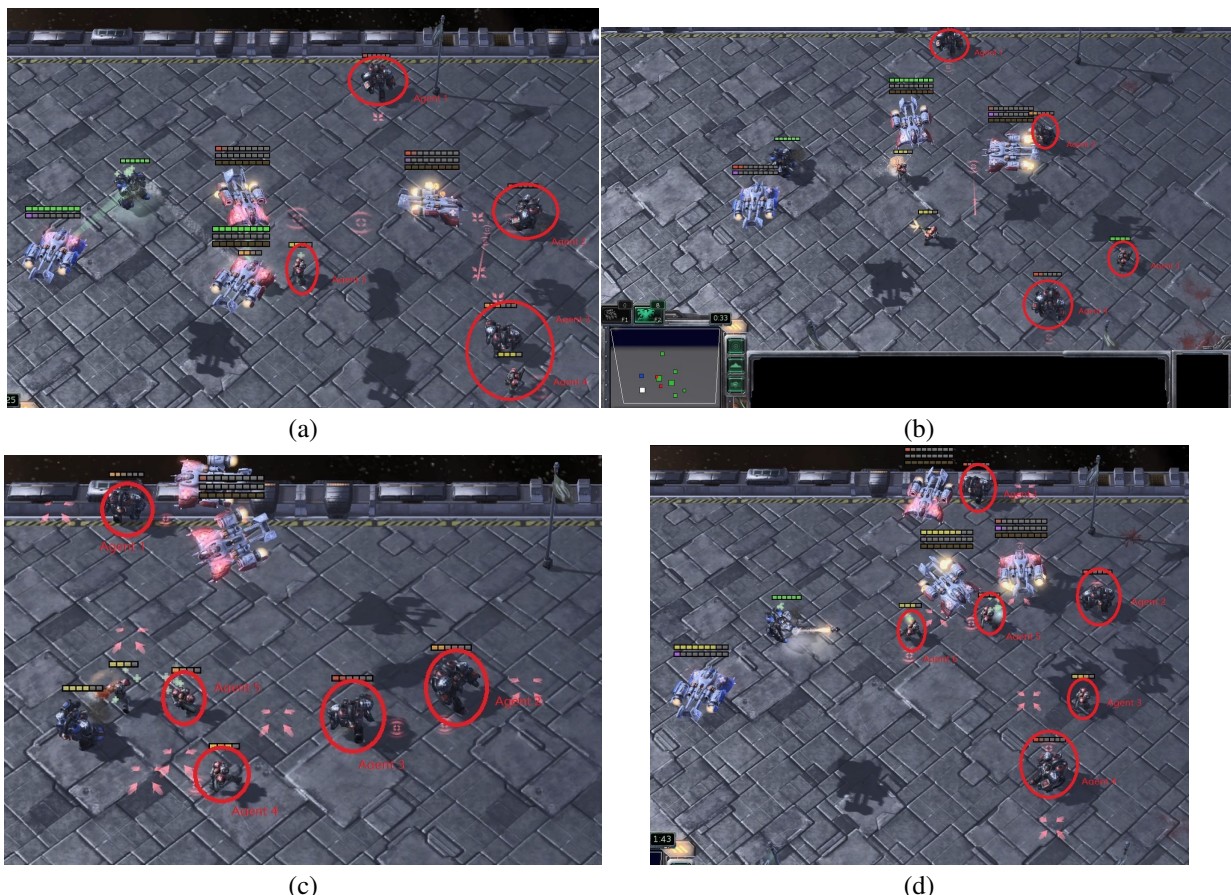

(a)

(b)

(c)

(d)

Figure 6: Visualization of wandering issue on Defense_3_Subtask. The result is obtained by the policy learned by QMIX. We marked the agent **with** the wandering issue with a red circle.

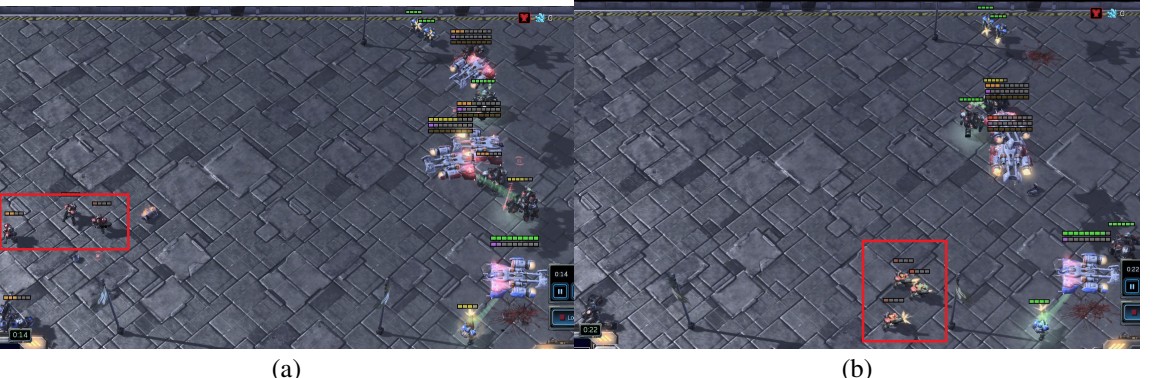

(a)

(b)

Figure 7: Visualization of Defense_3_Subtask. The result is obtained by the policy learned by e-QMIX. We marked the agent **without** the wandering issue with a red rectangle.

### A.3.3  ACTION SPACE

The discrete set of actions that agents are allowed to take consists of move[direction] (Four directions: north, south, east, or west), attack[enemy_id], stop and no-op. Dead agents can only take no-op action while live agents cannot. As healer units, Medivacs use heal[agent_id] actions instead of attack[enemy_id]. The maximum number of actions an agent can take ranges between 7 and 70, de-

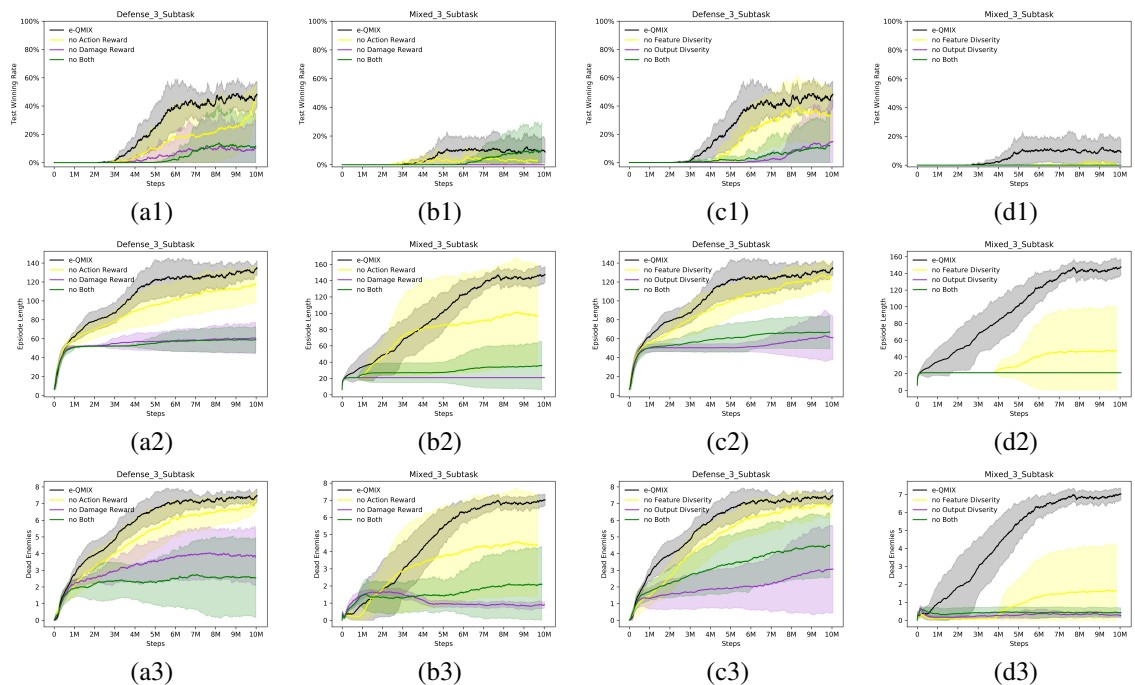

Figure 8: (a1-d1) Ablation studies on the test winning rate of e-QMIX. (a2-d2) Ablation studies on the episode length of e-QMIX. (a3-d3) Ablation studies on the number of dead enemies of e-QMIX.

pending on the scenario. To ensure decentralisation of the task, agents can use the attack[enemy_id] action only on enemies in their shooting range. This additionally constrains the ability of the units to use the built-in attack-move macro-actions on the enemies that are far away. We set the shooting range equal to 6 for all agents. Having a larger sight range than a shooting range forces agents to make use of the move commands before starting to fire.

### A.3.4 REWARDS

The overall goal is to maximise the winning rate for each battle scenario. The default setting is to use the shaped reward, which produces a reward based on the hit-point damage dealt and enemy units killed, together with a special bonus for winning the battle. The exact values and scales for each of these events can be configured using a range of flags. To produce fair comparisons we encourage using this default reward function for all scenarios. We also provide another sparse reward option, in which the reward is +1 for winning and -1 for losing an episode.

### A.4 EXPERIMENT SETTINGS

**Running Machine** We run all experiments on a computer with a CPU "AMD Ryzen 9 7900X 12-Core Processor", a GPU "Geforce RTX 3090", a 64G memory and a 1T storage space. The time it takes to run one run of each baseline on this machine is approximately 4-6 hours. We list the hyperparameters of the baselines used in the experiments in Table 5. The hyperparameters of e-QMIX remain the same as those of QMIX, and its additional hyperparameters are shown separately in Table 4.

**Source Code** To evaluate the effectiveness of current state-of-the-art (SOTA) methods on CTC tasks, we select eight representative MARL methods spanning three key categories: policy diversity,

Table 3: The maximum test winning rate of QMIX, GoMARL, ROMA and e-QMIX with RER on all CTC tasks across 5 seeds. D2S represent Defense_2_Subtask; D3S represent Defense_3_Subtask; D4S represent Defense_4_Subtask; M2S represent Mixed_2_Subtask; M3S represent Mixed_3_Subtask.

| Tasks | QMIX | GoMARL | ROMA | e-QMIX |
|-------|--------|---------|------|--------|
| D3S | 81.25% | 93.75% | 0 | 81.25& |
| D4S | 34.375% | 0 | 0 | 25% |
| M2S | 31.25% | 28.125% | 0 | 40.625% |
| M3S | 0 | 0 | 0 | 56.25% |

agent grouping, and hierarchical MARL. The source code of each baseline are: EOI[2], DCC[3], CDS[4], RODE[5], ROMA[6], GoMARL[7], LDSA[2], HSD[2] and QMIX[8].

## A.5 EXPERIMENT RESULTS SUPPLEMENT

In this section, we present supplementary experimental results related to both the benchmark baselines and the proposed guiding solution.

Regarding the guiding solution, Table 3 reports the maximum test winning rates of QMIX, Go-MARL, ROMA, and e-QMIX across five random seeds on each CTC task. Notably, e-QMIX achieves a non-zero maximum test winning rate on all four tasks, outperforming all other methods on the Mixed_2_Subtask and Mixed_3_Subtask. ROMA consistently fails to improve, maintaining a maximum test winning rate of zero across all tasks, indicating that RER has no positive impact on its performance. RER improves GoMARL's performance on the Defense_3_Subtask and Mixed_2_Subtask, while QMIX benefits from RER on all tasks except Mixed_3_Subtask.

Fig. 9 presents the test winning rate, episode length, and the number of dead enemy units for QMIX, GoMARL, ROMA, and e-QMIX across all CTC tasks, averaged over five seeds. The test winning rate analysis, discussed in the main content, supports the conclusion that **e-QMIX, under the guidance of RER, demonstrates the capability to achieve both division of labor (DOL) and cooperation across all CTC tasks.** This conclusion is further substantiated by examining the trends in episode length and the number of dead enemy units. e-QMIX exhibits a consistent increase in episode length across all tasks, far surpassing the thresholds of 7 and 21—indicators of successful DOL in both defense and mixed task settings. The corresponding increase in the number of dead enemies further reinforces this interpretation. When combined with e-QMIX's test winning rates, these findings collectively validate that e-QMIX, supported by RER, is capable of both effective division of labor and subsequent cooperation. Fig. 10 presents the test winning rate, episode length, and the number of dead enemy units for baselines with episode runner. The same conclusion can be drawn from these results.

## A.6 VISUALIZATION

We visualize the policies learned by QMIX on the defense_3_Subtask, as illustrated in Fig. 6. Agents exhibiting wandering behavior are highlighted with red circles. In Fig. 6(a), Agents 1–4 begin wandering after completing their assigned subtasks. Notably, Agent 5 starts wandering **before** completing its subtask. This premature deviation is likely caused by policy interference due to parameter sharing in QMIX, which leads to undesirable policy similarity across agents. In Fig. 6(b), Agents 1–4 exhibit wandering behavior post-subtask completion. However, in contrast to (a), no agent begins wandering prior to completing its assigned objective. A similar pattern is observed in Fig. 6(c),

[2]https://github.com/jiechuanjiang/eoi_pymarl

[3]https://github.com/chaobiubiu/DCC

[4]https://github.com/lich14/CDS

[5]https://github.com/TonghanWang/RODE

[6]https://github.com/TonghanWang/ROMA

[7]https://github.com/zyfsjycc/GoMARL

[8]https://github.com/hijkzzz/pymarl2

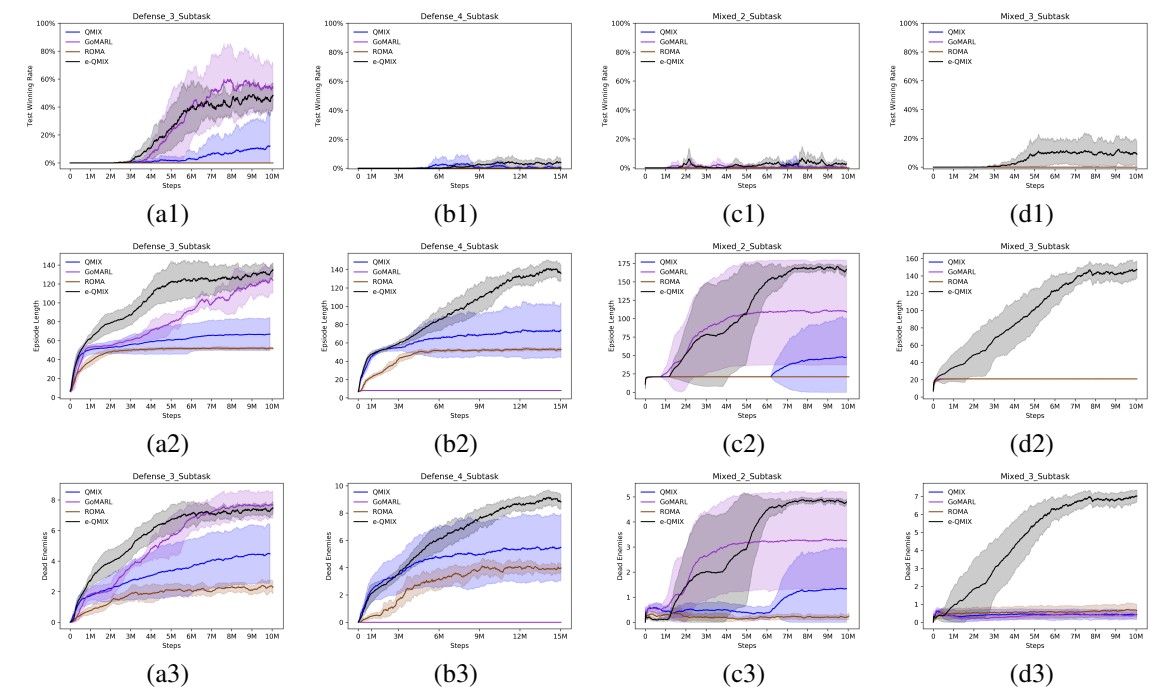

Figure 9: (a1-d1) Performance of baselines (parallel runner) on CTC tasks with RER. (a2-d2)Episode length of baselines on CTC tasks with RER. (a3-d3) The number of dead enemies of baselines on CTC tasks with RER.

where all agents begin wandering only after fulfilling their respective subtasks. Fig. 6(d) resembles the scenario in Fig. 6(a), where Agents 1–5 wander post-subtask, and Agent 6 begins wandering prematurely—again potentially due to policy interference. In summary, **wandering behavior predominantly emerges after agents complete their assigned subtasks**. While premature wandering (i.e., before subtask completion) is less frequent, it tends to follow the onset of wandering by other agents. This behavior significantly undermines coordinated performance and is a primary factor contributing to QMIX's persistent zero test winning rate on defense_3_Subtask. This conclusion is further supported by the episode length trends in Fig. 3(a1) and the test winning rate results in Fig. 2(a1).

Fig. 7 presents the policy learned by e-QMIX on the defense_3_Subtask. In Fig. 7(a), the three agents marked in red rectangle are initially assigned to a specific subtask. After successfully completing their assigned subtask, these agents proceed to another subtask to assist their teammates, as illustrated in Fig. 7(b). This cooperative behavior demonstrates that the combined design of e-QMIX and the RER effectively mitigates the wandering issue commonly observed in standard QMIX. The reduction in wandering behavior is a key factor contributing to the improved test winning rate of e-QMIX with RER on the defense_3_Subtask.

## A.7 ABLATION STUDY

We conduct ablation studies on both the RER (Rule-based External Reward) design and the e-QMIX architecture to investigate the contribution of their respective components to overall performance. Specifically, we examine the impact of removing the damage reward and action reward from RER, as well as the effects of removing the feature diversity and output diversity modules from e-QMIX. The results of these experiments are presented in Fig. 8.

As illustrated in Fig. 8(a1, b1), ablating either the damage reward or the action reward leads to a noticeable decline in e-QMIX's performance across all CTC tasks. Among the two, the removal of the damage reward results in more pronounced performance degradation compared to the action reward. Specifically, on the Mixed_3_Subtask, ablating the damage reward reduces the test win-

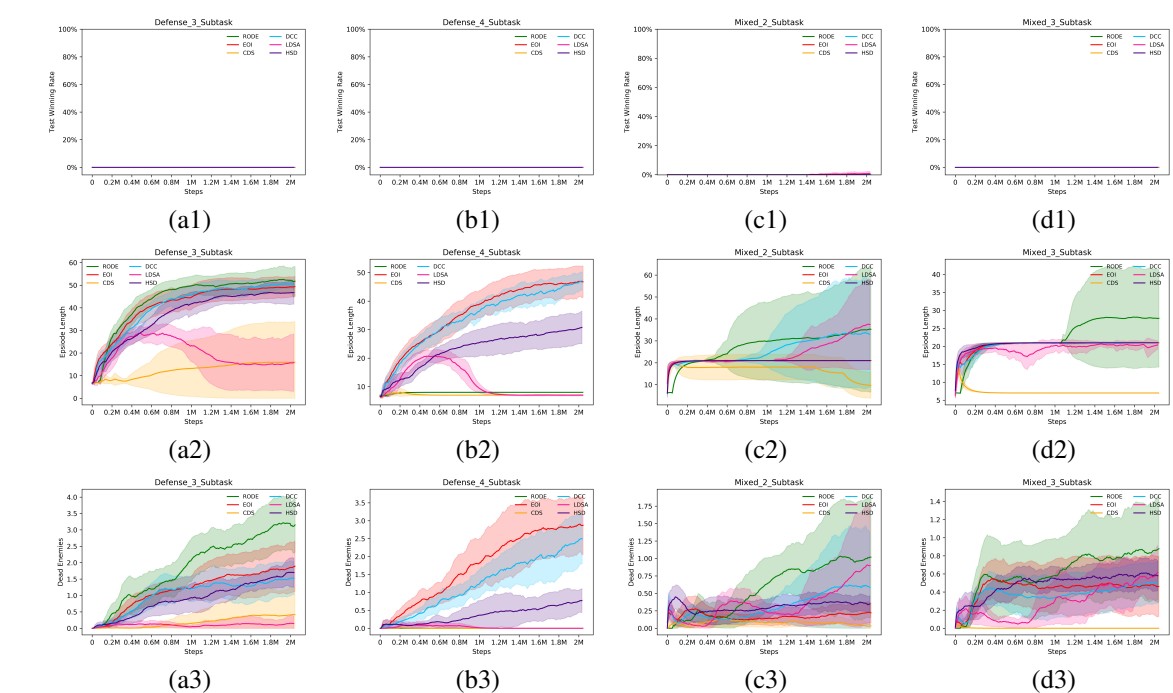

Figure 10: (a1-d1) Performance of baselines (episode runner) on CTC tasks with RER. (a2-d2)Episode length of baselines on CTC tasks with RER. (a3-d3) The number of dead enemies of baselines on CTC tasks with RER.

ning rate to zero, whereas ablating the action reward leads to a decrease in performance but still maintains a non-zero winning rate. Simultaneously removing both reward components causes a substantial performance drop on Defense_3_Subtask. Interestingly, on Mixed_3_Subtask, although the mean performance across 5 seeds declines, the maximum test winning rate increases. This suggests that e-QMIX may still develop effective DOL and cooperation policies even in the absence of explicit reward guidance. These results highlight that while the RER effectively guide agent behavior in many scenarios, their benefits are not universal across all task settings. This observation underscores the need to carefully consider task-specific dynamics when designing reward shaping. Further supporting this conclusion are the trends observed in episode length (Fig. 8(a2, b2)) and the number of defeated enemies (Fig. 8(a3, b3)). Both metrics exhibit consistent declines in response to reward ablations, mirroring the corresponding drop in test winning rates and reinforcing the critical role of RER in enhancing performance.

Similarly, as shown in Fig. 8(c1, d1), ablating either the feature diversity or output diversity module leads to a substantial decline in the performance of e-QMIX across the evaluated tasks. Among the two, the removal of output diversity has a more pronounced negative impact than the removal of feature diversity. On the Mixed_3_Subtask, for instance, eliminating the output diversity module reduces the test winning rate to zero, whereas removing feature diversity results in a performance drop but still maintains a non-zero winning rate. When both diversity modules are ablated, e-QMIX effectively reverts to the original QMIX architecture, which exhibits a test winning rate of zero on the Mixed_3_Subtask. Notably, on the Defense_3_Subtask, simultaneously removing both modules yields results similar to those observed when output diversity alone is removed, further highlighting the dominant influence of output diversity on overall performance. These findings underscore the critical importance of both diversity components in enabling e-QMIX to learn differentiated and effective agent policies. The corresponding declines in episode length (Fig. 8(c2, d2)) and the number of defeated enemies (Fig. 8(c3, d3)) provide additional support for this conclusion, as these reductions align closely with the observed decreases in test winning rate.

Table 4: Hyperparameters used for e-QMIX. Other hyperparameters is same with QMIX.

| | Defense_3_Subtask | Defense_4_Subtask | Mixed_2_Subtask | Mixed_3_Subtask |
|---|---|---|---|---|
| $\alpha$ Start | 1.0 | 2.0 | 1.0 | 2.0 |
| $\alpha$ Finish | 0.1 | 0.5 | 1.0 | 0.5 |
| $\alpha$ Anneal | 5000000 | 5000000 | 5000000 | 5000000 |
| $\beta$ | 0.1 | 0.1 | 0.0 | 0.1 |
| $\gamma$ | 0.1 | 0.1 | 0.1 | 0.1 |
| n_expert | 4 | 4 | 4 | 4 |
| topk | 2 | 2 | 2 | 2 |
| n_heads | 4 | 4 | 4 | 4 |
| rnn_dim | 64 | 48 | 64 | 64 |

Table 5: Hyperparameters used in our experiments. These are the values in the corresponding configuration file in the source code. Action Selectors used in all experiments are epsilon-greedy.

| | ROAM | RODE | CDS | EOI | QMIX | GoMARL | LDSA | DCC | HSD |
|---|---|---|---|---|---|---|---|---|---|
| Batch Size | 32 | 32 | 32 | 32 | 128 | 128 | 32 | 32 | 32 |
| parallel | 8 | 1 | 1 | 1 | 8 | 8 | 1 | 1 | 1 |
| Buffer Size | 5000 | 5000 | 5000 | 5000 | 5000 | 5000 | 5000 | 5000 | 5000 |
| beta | / | / | 0.07 | / | / | / | / | / | / |
| beta1 | / | / | 2.0 | / | / | / | / | / | / |
| beta2 | / | / | 1.0 | / | / | / | / | / | / |
| critic_lr | 0.0005 | 0.0005 | 0.0005 | 0.0005 | 0.0005 | 0.0005 | 0.0005 | 0.0005 | 0.0005 |
| double_q | true | true | true | true | / | / | true | true | true |
| $\epsilon$ Start | 1.0 | 1.0 | 1.0 | 1.0 | 1.0 | 1.0 | 1.0 | 1.0 | 1.0 |
| $\epsilon$ Finish | 0.05 | 0.05 | 0.05 | 0.05 | 0.05 | 0.05 | 0.05 | 0.05 | 0.05 |
| $\epsilon$ Anneal | 50 | 70000 | 500000 | 500000 | 100000 | 100000 | 500000 | 500000 | 500000 |
| gain | / | / | / | / | 0.01 | / | / | / | / |
| gamma | 0.99 | 0.99 | 0.99 | 0.99 | 0.99 | 0.99 | 0.99 | 0.99 | 0.99 |
| grad_clip | 10 | 10 | 10 | 10 | 10 | 10 | 10 | 10 | 10 |
| hyper_dim | / | 64 | 64 | 64 | 64 | 32 | 64 | 64 | 64 |
| hyper_layer | / | 2 | / | 2 | / | / | 2 | 2 | 2 |
| kl_weight | 0.0001 | / | / | / | / | / | / | / | / |
| latent_dim | 3 | / | / | / | / | / | / | / | / |
| lr | 0.0005 | 0.0005 | 0.0005 | 0.0005 | 0.0001 | 0.0001 | 0.0005 | 0.0005 | 0.0005 |
| load_step | 0 | 0 | / | 0 | / | 0 | 0 | 0 | 0 |
| log_interval | 10000 | 10000 | 10000 | 10000 | 10000 | 10000 | 10000 | 10000 | 10000 |
| mixing_dim | 32 | 32 | 32 | 32 | 32 | 32 | 32 | 32 | 32 |
| num_circle | / | / | 2 | / | / | / | / | / | / |
| num_kernel | / | / | 4 | / | / | / | / | / | / |
| n_subtasks | / | / | / | / | / | / | 4 | / | / |
| obs_agent_id | true | true | false | true | true | true | true | true | true |
| obs_last_act | true | true | true | true | true | true | true | true | true |
| optim_alpha | 0.99 | 0.99 | 0.99 | 0.99 | 0.99 | 0.99 | 0.99 | 0.99 | 0.99 |
| optim_eps | 1e-05 | 1e-05 | 1e-05 | 1e-05 | 1e-05 | 1e-05 | 1e-05 | 1e-05 | 1e-05 |
| rnn_dim | 64 | 64 | 64 | 64 | 64 | 64 | 64 | 64 | 64 |
| Runner | parallel | episode | episode | episode | parallel | parallel | episode | episode | episode |
| $t_{max}$ | 10050000 | 2050000 | 2050000 | 2050000 | 10050000 | 10050000 | 2050000 | 2050000 | 2050000 |
| test_interval | 20000 | 10000 | 10000 | 10000 | 10000 | 10000 | 10000 | 10000 | 10000 |
| test_episode | 24 | 32 | 32 | 32 | 32 | 32 | 32 | 32 | 32 |
| var_floor | 0.002 | / | / | / | / | / | / | 0.002 | 0.002 |

In summary, the ablation results confirm that both components of the RER reward design and the architectural enhancements in e-QMIX are essential for achieving strong performance on the challenging CTC tasks.

