# OpenReview forum: "CTC: The Composite Task Challenge for Cooperative Multi-Agent Reinforcement Learning"
_ICLR.cc/2026/Conference — Submitted to ICLR 2026_

### Official Review · Reviewer_kuJ6 · 2025-10-20

**Soundness:** 3
**Presentation:** 2
**Contribution:** 2
**Rating:** 4
**Confidence:** 3

**Summary:**

The authors propose CTC (Composite Task Challenge), consisting of a suite of tasks designed to appropriately evaluate agents ability in cooperation and division of labor. They evaluate several existing methods in their benchmark, and demonstrate the limitation of those methods. To demonstrate the solvability of their CTC tasks, the authors manually design external rewards and modify the network structure of QMIX to leverage it, which enables improved performance.

**Strengths:**

The authors' motivation is reasonable: DOL is an important topic and is closely related to cooperation efficiency, while existing MARL testbeds do not explicitly evaluate this feature.

The two principles for CTC task designing is reasonable, and the proposed CTC tasks indeed align with them.

Besides the benchmark, the authors contribute a guiding solution to demonstrate that the proposed task is solvable.

**Weaknesses:**

1. I found there is much space to improve the paper writing.
* the label/title fonts in the figures for experiment results are too small, I have to zoom in to read it. and same issue for Figure 1.
* line 191 "Fig. (d)-(e)", where is (e)?
* For figure 2 and 3, if I understand correctly, both "QMIX, GoMARL, ROMA" and "RODE, DCC, EOI, LSDA, CDS, HSD" are methods, then why plot them separately in two pictures?

    Besides, I think the author need to explain it clearly about what is the state (or observation) space, action space and transition function for the proposed CTC tasks. I did not find a clear explanation in the main text, and I think that is essential to understand the challenges of the tasks and the results.

    In addition, usually, a preliminary section should be included, introducting notation and providing a formal definition for the multi-agent cooperative games.

2. Although I believe the proposed CTC are valid contributions to this community, the overall contribution of this paper does not reach the bar of acceptance of this top-tier conference. The main contribution is still restricted in proposing a benchmark, but does not contribute effective solutions to this challenge.

    Given the limited performance improvement in Figure 5, I'm not convinced that the external reward method is effective in resolve this challenge.

**Questions:**

What are the state and action spaces for the proposed CTC tasks, and what are the state-action transition function?

---

### Official Review · Reviewer_mHph · 2025-10-31

**Soundness:** 2
**Presentation:** 2
**Contribution:** 1
**Rating:** 2
**Confidence:** 4

**Summary:**

This paper identifies a perceived gap in cooperative Multi-Agent Reinforcement Learning (MARL) benchmarks, arguing that existing tasks do not explicitly require a division of labor (DOL). To address this, the authors propose the Composite Tasks Challenge (CTC), a suite of environments where success is contingent upon both DOL and cooperation, as failure in any subtask leads to overall failure. The paper evaluates several existing MARL methods on these tasks, reports uniformly poor performance, and demonstrates the tasks' solvability with a guiding solution. The work aims to establish CTC as a new benchmark for advancing cooperative MARL.

**Strengths:**

The paper is written in a clear and concise manner, making its core proposition and the design principles of the CTC easy to understand.

**Weaknesses:**

1. The paper's foundational motivation is potentially narrow and the contributions are limited. DOL in MARL is one approach, among others, to tackle core challenges like combinatorial action spaces and credit assignment. The claim that the absence of benchmarks explicitly requiring DOL is a critical gap is not sufficiently justified. The paper would benefit from a deeper theoretical discussion on why DOL is a necessary focus for future benchmarks, rather than an emergent property, to elevate its perceived research value and technical depth.

2. There are some issues with figure and table presentation, as well as some logic Flows. The organization and referencing of figures and tables are confusing and hinder readability. For instance, Table 2 is referenced before Table 1 in Section 2.2. All tables are placed in the appendix without clear justification. Figure 1 is introduced piecemeal, with part (c) being discussed in Section 2.1 before the figure is fully explained in Section 2.2. The authors should thoroughly restructure Section 2 to ensure a logical narrative flow. Figures and tables should be referenced in a sequential and intuitive manner, and key tables (especially those with results) should be integrated into the main text.

2. The empirical evaluation is insufficient and outdated, and lacks persuasiveness. The selection of "representative methods" is heavily skewed towards older algorithms, with only one method (GoMARL) from the last three years. To robustly demonstrate the challenge posed by the CTC benchmark, it is crucial to include and test against a wider range of recent state-of-the-art cooperative MARL methods. The current selection limits the claim about the benchmark's necessity and difficulty.

3. The use of space in the main text is insufficient. Figures 2 and 5, as currently presented, consume valuable space in the main text without providing commensurate informational value. This contributes to the paper feeling somewhat thin. It is recommended that these figures be moved to the appendix to free up space for more critical content, such as the key tables from the appendix and a more detailed discussion of the methodology and results.

4. The proposed guiding solution appears to be a handcrafted, scenario-specific heuristic. Its generality and scalability are highly questionable, as evidenced by its inconsistent performance. It shows a notable improvement only in the Defense_3_Subtask while having minimal impact on others. Furthermore, the paper fails to provide any in-depth analysis explaining why the guiding solution helps certain methods like GoMARL but not others. This lack of analysis significantly weakens the insights drawn from the solution and its value to the research community.

**Questions:**

Please refer to the above weaknesses.

---

### Official Review · Reviewer_diEv · 2025-11-08

**Soundness:** 2
**Presentation:** 2
**Contribution:** 2
**Rating:** 2
**Confidence:** 3

**Summary:**

The paper modifies SMAC to probe division of labour in cooperative MARL, defining it as inter-agent functional specialisation on asymmetric subtasks with complementary behaviours. The authors argue that existing benchmarks lack learnable roles and explicit penalties for subtask failure, particularly under task asymmetry and agent heterogeneity. Empirically, standard MARL methods fail to learn on the proposed composite tasks, yielding near-zero average win rates. To address this, the authors introduce a deeper QMIX variant augmented with an auxiliary diversity-promoting reward to encourage differentiated policies across agents, which produces modest performance on one of their tasks.

**Strengths:**

* The focus on emergent division of labour, where agents autonomously allocating complementary subtasks in cooperative settings, is an interesting extension that pushes beyond standard coordination-only objectives.


* Broad benchmarking across multiple MARL algorithms is valuable to the community and non-trivial to execute, given nonstationarity and the resulting instability of learned policies and evaluations.

* Using SMAC's framework and environment facilitates comparisons to other MARL algorithms that have used SMAC

**Weaknesses:**

* The near-zero win-rate plots can be read by non-MARL reviewers as implementation or hyperparameter-tuning failure rather than inherent task difficulty.


* There is no behavioural evidence or standard learning curves (e.g., cumulative return) to show agents are learning or even trying to learn. SMAC exposes rich metrics (like unit health, shots fired, damage dealt, etc.) that should be plotted. Episode length alone is ambiguous in SMAC (it can reflect fast deaths or prolonged avoidance). A stronger metric is to timestamp when each subtask is completed, even if that requires modifying SMAC.


* The baselines rely on QMIX variants but omit stronger, commonly used methods such as MAPPO/IPPO.


* The rule-based reward shaping is interesting methodologically, but the “damage” reward appears contrived and not clearly related to division of labour.


* The wandering behaviour resembles the lazy-agent problem, which has been addressed for QMIX via intrinsic rewards (Liu et al., ICML 2023): https://proceedings.mlr.press/v202/liu23ac.html


* It is unclear whether the new reward function’s alpha and beta coefficients are learnable; they seem fixed with a decay, which confounds the results.


* The proposed method still fails on the introduced tasks; if the role-encouraging reward (RER) fails, it is hard to regard it as a solution.


* QMIX’s additive value factorization pushes agents toward a single coordinated objective and may be a poor fit for persistent role specialization.


* The problem framing is not especially novel; multi-task RL has a long history (https://arxiv.org/abs/1609.07088, https://arxiv.org/abs/1611.01796) and related perspectives in MARL (https://arxiv.org/abs/1703.06182) should be engaged more directly.


* SMACv2 (https://arxiv.org/abs/2212.07489) is not used; results on SMACv2 would improve relevance and comparability.

**Questions:**

* Why not include a qualitative analysis of behaviour in the style of Hierarchical Cooperative MARL with Skill Discovery https://arxiv.org/abs/1912.03558 or The Emergence of Individuality https://arxiv.org/abs/2006.05842 ? This would demonstrate that the algorithms are actually learning and that distinct roles emerge, beyond what win rates alone show.

* Why not pretrain on standard SMAC and then fine-tune on the CTC tasks to quantify how CTC differs from (and adds to) SMAC? A transfer experiment would directly test the benchmark’s necessity by measuring gains or gaps in sample efficiency, asymptotic performance, and subtask completion.

* Intrinsic-reward models seem well-suited to the author's failure modes, especially rewards that estimate an agent’s causal influence on team progress (e.g., Liu et al., ICML 2023: https://proceedings.mlr.press/v202/liu23ac.html). Given that you already add a rule-based shaping term, why not use intrinsic rewards (or learn the shaping coefficients) instead of hand-tuned bonuses?

* Why evaluate only derivatives of QMIX and not closely related architectures like QTRAN (https://arxiv.org/abs/1905.05408) or MAVEN (https://arxiv.org/abs/1910.07483)? These offer greater expressivity and structured exploration, and would clarify whether the observed failures are specific to QMIX’s factorization or persist across stronger baselines.

---

### Official Review · Reviewer_AFYS · 2025-11-08

**Soundness:** 3
**Presentation:** 2
**Contribution:** 2
**Rating:** 4
**Confidence:** 3

**Summary:**

This paper introduces a benchmark named Composite Tasks Challenge (CTC), a set of cooperative MARL scenarios explicitly constructed so that division of labor (DOL) is necessary for success and failure on any atomic “subtask” causes overall failure. The tasks are instantiated in SMAC with pursuit/defense subtasks that start simultaneously and are spatially disjoint, which prevents a single agent from solving multiple subtasks at once.

**Strengths:**

This testbed is an interesting addition to the very much stale MARL benchmark landscape. The design principle is well-motivated, and the tasks certainly seems hard for current methods. RER and the e-QMIX is not positioned as “the answer” but as proof of solvability and a diagnostic tool. The authors also provided implementation descriptions, environment details, and hyperparameters.

**Weaknesses:**

I do not feel comfortable about the writing style. Personally I think this paper have too much repetitive narrative, the motivations are iterated over and over, the overall narrative could really use some further de-LLM writing. The Overcook is even introduced twice in the appendix.

For e-QMIX, the output diversity module uses biases derived from the concatenation of all agents’ observations and a multi-head attention over other agents’ histories, this is a CTDE violation. If they are used at test-time, please either justify this design choice versus CTDE, or revise the architecture to keep per-agent policies decentralized.

**Questions:**

How does performance and compute scale as the number of simultaneous subtasks grows beyond 4?
Since runner types affect the semantics of “steps,” consider reporting environment frames or episodes to normalize across baselines?

---

### Official Review · Reviewer_BNko · 2025-11-09

**Soundness:** 2
**Presentation:** 1
**Contribution:** 2
**Rating:** 2
**Confidence:** 4

**Summary:**

CTC introduces a new benchmark for evaluating algorithms' ability to divide labor among agents. It is based on the StarCraft Multi-Agent Challenge benchmark. Namely, CTC uses modular subtasks which cannot all be solved by a single agent, thereby requiring good division of labor to succeed. The authors evaluate on eight common MARL methods and find that none of them are able to solve the tasks. By introducing a simple reward function and a few modifications to QMIX, they see a very high success rate.

**Strengths:**

The paper identifies an important capability that many MARL benchmarks currently do not test for. The authors test a wide range of existing methods on their benchmark to check the validity. Additionally, the authors provide a strong solution for their benchmark using a hand-designed reward. The paper raises interesting questions about how existing algorithms learn division of labor behavior, and how future work may evaluate this capability.

**Weaknesses:**

- If the goal of the paper is to introduce a challenging new benchmark, it is unclear that it has succeeded. The paper demonstrates that with a simple hand-designed reward and some modifications to QMIX, a policy can achieve a very high success rate on the new benchmark. It is unclear what remains to be done on this benchmark to see further improvements.
- The paper lacks critical details that would greatly improve readability. A few suggestions:
    - Please give a comprehensive description of the starcraft environment. Please specify the (PO)MDP.
    - Please make it clear what the opponent policies are
        - Please add more discussion in the captions beneath the figures.
        - Please improve the overall clarity of the figures. I am not sure what Figure 1 is conveying. Is this a map? Who are the agents? Do the physical locations in the figure correspond to locations on the map?
        - Please include the important fact that this is based on the starcraft benchmark in the abstract, in the introduction, and in all other relevant sections. This is very important.
- In section 4.1, you use the notation r^t. I assume you are referring to the reward at timestep t, not the reward raised to the power of t. It may be better to use r_t, which is more standard.
- I would suggest using fewer acronyms. It would improve readability.
- I would also suggest reworking the discussion around how you constructed the CTC benchmark. A better flow may be: Here is the starcraft environment which is multi-agent, we can choose the map in the starcraft environment to specifically test division of labor, this environment has this POMDP, we choose to place "subtasks" randomly around the environment, these are far enough apart that each agent has to choose a single task to try to accomplish, and then here are the subtasks.
- It is very important that you add a solid discussion of why this benchmark is still valuable, even though you have an alright solution for it at the moment. You may be able to make this case, but it must at least be included.

**Questions:**

- Did you perform hyperparameter sweeps on the baselines?
    - What is the reward the baselines are trained on?
        - Do the policies share parameters?
        - What is the objective you are optimizing? You evaluate a good number of baselines, so it might suffice to discuss QMIX in more depth as this is the algorithm you build on.
        - Are the subtasks in the same map? So, for example, could any agent feasibly attempt to work on any subtask? Or are they physically separated?
- Would you please explain lines 333-338? I am unsure whether these "behavioral features" are features of a particular opponent policy (what opponent?) or behavioral features that you believe are desirable in Starcraft?
- Did you try using other dense rewards?
- Did you try using your dense reward with the other baselines? How did they perform?
- In equation 2, are you feeding this total reward to a single policy? In that case, is this not a MARL training because you have a single policy you are optimizing with RL? I may be missing something.
- Why is Overcooked not a good environment for this type of evaluation? I believe Overcooked has division of labor tasks as well.

---

### Meta-Review · Area_Chair_WPL8 · 2026-01-04

**Summary:**

This paper introduces the Composite Task Challenge (CTC), a new benchmark designed to require explicit division of labor and cooperation in multi-agent reinforcement learning. It evaluates several existing MARL methods on CTC, showing their failure, and provides a proof-of-concept solution to demonstrate task solvability.

However, Reviewers BNko and mHph argued that the benchmark's core challenge is undermined, as the authors' own simple, hand-designed solution achieves high success rates, questioning what fundamental problem remains. Reviewers mHph and diEv noted that the empirical evaluation is insufficient, relying on outdated baselines and lacking compelling behavioral analysis or key metrics. Reviewers BNko, AFYS, and kuJ6 highlighted serious presentation issues, including unclear writing, poor organization of figures/tables, and missing critical details about the environment. Reviewers mHph and kuJ6 concluded that the overall contribution is incremental and does not meet the threshold for a top-tier conference publication.

Based on the above viewpoint, I recommend a reject decision.

**Reviewer Concerns:**

This paper did not undergo a rebuttal process.

**Reviewer Scores:**

Since the authors did not submit a rebuttal, the scores should remain unchanged.

---

### Decision · Program_Chairs · 2026-01-26

Reject